# Identification of Mudanjiang Phlebovirus in the Daxing’anling Region of China

**DOI:** 10.3390/v15122353

**Published:** 2023-11-30

**Authors:** Chenli Fang, Yantao Liu, Fang Tang, Liming Liu, Peijun Guo, Yunfa Zhang, Jingtao Zhang, Xiaofang Ma, Zhenyu Hu, Shuang Li, Gang Wang, Shiwei Cheng, Xingxiao Zhang, Jianlong Zhang, Xiaoai Zhang, Wei Liu

**Affiliations:** 1College of Life Science, Ludong University, Yantai 264025, China; fcl1285181262@163.com (C.F.); swcheng@ldu.edu.cn (S.C.); zhangxingxiao@ldu.edu.cn (X.Z.); 2State Key Laboratory of Pathogen and Biosecurity, Beijing Institute of Microbiology and Epidemiology, Beijing 100071, China; fireflyzyf@163.com (Y.Z.); zjt66882022@yeah.net (J.Z.); zx2577102876@163.com (S.L.); wangg0304@139.com (G.W.); 3Qingdao Municipal Center for Disease Control and Prevention, Qingdao 266033, China; 18853280878@163.com (Y.L.); xf_mal@163.com (X.M.); 4Institute of Medical Prevention and Control of Public Health Emergencies, Characteristic Medical Center of the Chinese People’s Armed Police Force, Beijing 102613, China; tf4065@163.com; 5College of Animal Science and Technology, Jilin Agricultural Science and Technology University, Jilin 132101, China; liuliming@jlnku.edu.cn; 6Yantai Center for Disease Control and Prevention, Yantai 264003, China; cdcguo@126.com; 7School of Public Health, Anhui Medical University, Hefei 230032, China; h19980706@163.com

**Keywords:** tick-borne phlebovirus, mudanjiang phlebovirus, *Ixodes persulcatus*, *Meriones meridianus*, China

## Abstract

Mudanjiang phlebovirus (MJPV) is a newly discovered phlebovirus, initially detected from *Ixodes persulcatus* ticks in China in 2022. In this study, by next-generation sequencing (NGS) on a wide variety of ticks and wild small animals in China, we detected MJPV from *I. persulcatus* and *Meriones meridianus*. Additionally, we conducted RT-PCR and sequencing on 1815 adult ticks and 805 wild small mammals collected from eight provinces in China between 2017 and 2021. MJPV RNA-positive results were found in 0.22% (4/1815) of tick samples, as well as in 0.12% (1/805) of rodent samples. All positive detections were obtained from Heilongjiang and Inner Mongolia. Sequencing analysis revealed nucleotide similarities ranging from 98.23% to 99.11%, as well as amino acid similarities ranging from 99.12% to100%, between the current MJPV strain and previously reported strains of MJPV. Phylogenetic tree analysis demonstrated that the previously reported MJPV strain along with our two variants clustered together with other tick-borne phenuiviruses, indicating their close relationship within this viral group. This study represents the first detection of MJPV infection in wild rodents, expanding the known host range for this virus in the endemic regions.

## 1. Introduction

As the primary hematophagous ectoparasites of vertebrates, ticks pose a significant global threat to public health. They serve as important vector hosts for a wide range of pathogens, including bacteria, fungi, protozoa, and viral pathogens that cause diseases in humans and animals worldwide [1]. Notably, tick-borne viral diseases (TBVDs) such as Crimean–Congo hemorrhagic fever (CCHF), tick-borne encephalitis (TBE), and severe fever with thrombocytopenia syndrome (SFTS) represent emerging or re-emerging infectious diseases that severely endanger human health [2]. In recent years, TBVDs have increasingly become a global public health concern due to the frequent discovery of new viral pathogens across expanding geographic ranges [3].

Phlebovirus is a genus of RNA viruses belonging to the family *Phenuiviridae* in the order Bunyavirales. According to the latest classification by the International Committee on Taxonomy of Viruses (ICTV), over 60 species have been identified within this genus. Most members of the *Phlebovirus* genus are arthropod-transmitted viruses known to cause diseases in humans and animals [4]. The most significant phlebovirus with implications for public health and veterinary importance is the Rift Valley fever virus (RVFV) [5]. Due to its ability to cause severe and fatal disease in humans, RVFV has been listed as a priority pathogen by the World Health Organization (WHO) [6]. Since the 2000s, novel phleboviruses have been continuously identified and classified based on their genetic and serological characterization [7,8,9]. In 2018, the Mukawa virus (MKWV), a newly discovered member of the genus *Phlebovirus*, was detected and isolated from Ixodes persulcatus ticks collected in Hokkaido, Japan [7]. A subsequent serological survey conducted among wildlife revealed the neutralizing antibodies against MKWV present in both Yezo deer and raccoons captured at the initial discovery site in Hokkaido [8]. Additionally, in Hokkaido, the Kuriyama virus (KURV), another virus belonging to the *Phlebovirus* genus, was firstly detected and isolated from *I. persulcatus* [8]. Our own group’s study, as well as others’, confirmed the presence of MKWV in *I. persulcatus* and *Haemaphysalis concinna* in northern China [10]. Recently, another new species within the *Phlebovirus* genus, the Mudanjiang phlebovirus (MJPV), was identified through next-generation sequencing (NGS) from *I. persulcatus* in Mudanjiang city, Heilongjiang province, China [9]. Given that *Phlebovirus* members have a wide geographic distribution across East Asia, we aimed to determine molecular evidence of MJPV by studying a diverse range of ticks and wild small mammals across an expanded geographic area in China.

## 2. Materials and Methods

### 2.1. Study Sites and Sample Collection

Free-living ticks were collected using a drag-flag method from flagging vegetation in the Inner Mongolia, Heilongjiang, Henan, Shanxi, Liaoning, Shandong, and Zhejiang provinces of China. The ticks were identified to the species level based on morphological characteristics and further confirmed by sequencing the mitochondrial 16S rDNA gene [11]. All tick samples were stored in tubes and placed in liquid nitrogen upon capture, transferred to the laboratory and stored at −80 °C for later use. Wild small mammals were captured with snap traps in Heilongjiang, Inner Mongolia, Xinjiang, and Henan provinces (or autonomous regions). They were identified by trained field biologists and further confirmed by sequencing the mitochondrial cytochrome b gene [12]. Organ specimens such as heart, lung, liver, kidney, and spleen of wild small mammals were obtained in a BSL-2 laboratory. All collected organ samples were stored in tubes at −80 °C prior to use.

### 2.2. Extraction of Viral RNA and Metagenomic Analysis by NGS

Seventy-five tick samples belonging to five species (*I. persulcatus*, *Dermacentor sivarum*, *Haemaphysalis longicornis*, *Rhipicephalus microplus*, and *H. concinna*) along with twelve spleen samples belonging to four dominant wild small animal species (*Meriones unguiculatus*, *Mus musculus*, *Rhombomys opimus*, and *Meriones meridianus*) underwent metagenomic analysis by NGS as previously described [13]. Briefly, total viral nucleic acids were extracted using AllPrep DNA/RNA Mini Kit (Qiagen), followed by depletion of rRNA using the MGIEasy rRNA Depletion Kit (BGI, China). A high-throughput sequencing library was constructed using the MGIEasy RNA Library Prep Kit (BGI) before being sequenced on the MGI2000 platform (BGI) to generate 150 bp paired-end reads. After processing the original data by filtering, trimming, and error removal to obtain high-quality reads, the reads were mapped to a reference genome sequence using BWA (Version: 0.7.15). The remaining reads were aligned against non-redundant bacterial, virus, fungal, and parasite databases also applying BWA. De novo assembly was performed using MEGAHIT (v1.1.2) software. The assembled contigs were blasted against the corresponding reference genome sequences to identify virus-related contigs.

### 2.3. Reverse Transcription-PCR (RT-PCR) and Sequencing

RT-PCR was conducted on 1815 adult ticks and 805 wild small mammals. Total nucleic acid was extracted using AllPrep DNA/RNA Mini Kit (Qiagen) from tick and tissue samples following manufacturer’s instructions. A one-step RT-PCR system was employed to test the extracted RNAs using two universal primers sets (ppL1 and ppL2) for the detection of tick-borne phleboviruses (TBPVs) as previously reported (Table 1) [14]. PCR amplification was performed using the PrimeScript™ one-step RT-PCR kit version 2 (TaKaRa) following the manufacturer’s instructions, with the following program: incubation at 50 °C for 30 min; initial denaturation at 94 °C for 2 min; followed by 40 cycles of denaturation at 94 °C for 30 s, annealing at 55 °C for 30 s, extension at 72 °C for 30 s; and final extension at 72 °C for 5 min. The amplified products were determined by agarose gel electrophoresis, followed by Sanger sequencing. All PCR tests were conducted in parallel with positive control (RNA from positive sample) and negative control (RNase-free water).

### 2.4. Phylogenetic Analysis

The nucleotide and amino acid sequences obtained from both currently detected MJPVs and other representative species from family *Phenuiviridae* downloaded from GenBank were aligned by ClustalW method using MEGA-X. Phylogenetic trees were constructed based on the maximum likelihood (ML) method, with the robustness of each node tested through bootstrap replications performed up to 1000 times.

### 2.5. S Nucleotide Sequence Accession Numbers

The MJPV sequences generated in this study were submitted to GenBank under the accession numbers OR723773-OR723775 (full-length), OR723777-OR723783 (partial L segment).

## 3. Results

### 3.1. Identification of MJPV in I. persulcatus and M. meridianus by NGS

The metagenomic analysis revealed the presence of MJPV-specific sequences in *I. persulcatus* ticks collected from Heilongjiang province in 2021, among a total of 252,079,800 reads obtained, with an annotation rate of 2.15% to MJPV. By analyzing 5,413,802 reads, we successfully obtained the complete sequences of the large segment (6424 bp), medium (M) segment (3330 bp), small (S) segment (1882 bp) of MJPV (strain HLJ-26), which have been deposited in GenBank under accession numbers OR723773-OR723775. However, no MJPV-specific sequence was detected in the other four tick species including *D. silvarum*, *H. longicornis*, *R. microplus*, and *H. concinna* using NGS technology.

Regarding wild small animal species subjected to metagenomic analysis, we identified MJPV-specific sequences from the spleen samples of *M. meridianus* captured in the Inner Mongolia Autonomous Region in 2019. Out of a total of 147,636,597 reads obtained, only 1049 were annotated as MJPV-related sequences with partial length information for the L segment (2844 bp; strain IM-54). This partial sequence was also deposited in GenBank under accession number OR723784. However, no specific sequence related to MJPV was found among the three other wild small animal species, including *M. unguiculatus*, *M. musculus*, and *R. opimus* by NGS.

### 3.2. Epidemiologic Investigation of MJPV in in Ticks and Wild Small Mammals

A total of 1815 adult ticks, comprising five tick species, and 805 wild small mammals, consisting of 37 rodent species, collected from September 2017 to October 2021, were tested for the presence of MJPV (Figure 1). The ticks were collected from seven provinces (Heilongjiang, Inner Mongolia, Henan, Shanxi, Liaoning, Shandong, and Zhejiang). Among them, *I. persulcatus* had the highest number of samples (*n* = 665), followed by *H. longicornis* (*n* = 599), *D. silvarum* (*n* = 336), *R. microplus* (*n* = 170), and *H. concinna* (*n* = 45). The wild small mammals were captured from four provinces (Heilongjiang, Inner Mongolia, Henan, and Xinjiang) (Figure 1). The most abundant rodent species included *M. unguiculatus* (*n* = 183), *M. musculus* (*n* = 105), *R. opimus* (*n* = 77), and *M. meridianus* (*n* = 67), all belonging to the *Muridae* family within the Rodentia order.

RT-PCR screening revealed that only a small portion (0.22%, 4/1805) of tick extracts yielded positive results for PCR amplification. All positive results were derived from *I. persulcatus* in Heilongjiang province (0.60%, 4/665), while no positive detection was found in other tick species, even at the same location. Furthermore, in terms of detection among wild small mammals, a single positive result was obtained from *M. meridianus* in Inner Mongolia out of all the 805 screened samples (Table 2).

### 3.3. Genome Characterization and Phylogenetic Analysis of MJPV

Based on the MJPV sequences in this study (accession numbers OR723773-OR723775) and previous report (accession numbers: ON408132.1, ON408133.1 and ON408134.1) [9], we conducted a pairwise similarity analysis. The nucleotide acid identity for full L, M, and S sequences was found to be 99.11%, 98.23%, and 98.73%, respectively, when compared with the known sequences (Appendix A). Amino acid analysis demonstrated that our MJPV was more closely related to the previously reported MJPV (aa similarity: RNA-dependent RNA polymerase, 99.67%; glycoprotein precursor, 99.43%; nucleocapsid protein, 100%; nonstructural protein, 99.12%) than to other genogroups of phleboviruses, which exhibited aa similarities ranging from 56.18 to 88.75% (Appendix A, Appendix A). The pairwise nucleotide and amino acid identities between our MJPV strains isolated from *I. persulcatus* and *M. meridianus* were determined as 99.40 and 100%, respectively (Appendix A). A phylogenetic tree was further constructed, which demonstrated that our two MJPV strains along with previously reported MJPV strains, KURV. and MKWV from Japan formed a clade together (Figure 2). All branches derived from the complete L, S and M segments of these viruses fell into the tick-borne phenuiviruses group in the phylogenetic analysis (Figure 3). Similar clustering patterns were observed based on amino acid sequences of the RNA-dependent RNA polymerase, glycoprotein precursor, nucleocapsid protein, and nonstructural protein (Figure 4). Collectively, these results indicate that the current virus represents a novel variant of MJPV.

## 4. Discussion

As a region characterized by its environmental richness and megadiversity [15], the Daxing’anling mountains and their surrounding areas harbor various environmental and sociodemographic factors that render them highly susceptible to the emergence and establishment of ticks as well as epidemic tick-borne pathogens. Over the last three decades, more than 30 tick-borne infections have been identified in this region. In this study, through extensive screening of a diverse range of ticks and dominant wild small mammals across eight provinces in China, we have determined the presence of MJPV in *I. persulcatus* from Heilongjiang province and *M. meridianus* from Inner Mongolia, respectively. Although with a low prevalence, this result once again emphasizes the need to focus on *I. persulcatus*, as a competent vector for emerging tick-borne diseases. Widely distributed in Asia and Eastern Europe, *I. persulcatus* can feed on 46 different host species while harboring 51 tick-borne agents [16]. The extensive positive detection results suggest an important role played by *I. persulcatus* in carrying phleboviruses such as MKWV, KURV, and MJPV. Notably, our study represents the first positive detection of MJPV in rodent species which implies that wild small animals may potentially act as reservoir or amplifying hosts within the transmission cycles of these newly identified phleboviruses. These findings significantly contribute to expanding our current understanding of the host characteristics, biogeography patterns, and genetic divergence associated with MJPV.

## Figures and Tables

**Figure 1 viruses-15-02353-f001:**
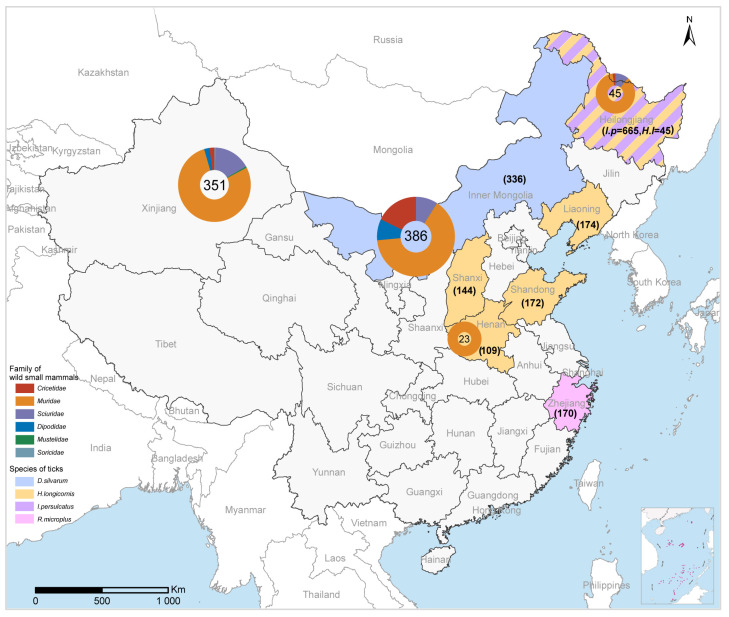
The collection sites for ticks and wild small mammals for MJPV detection. The sampling numbers for each family of wild small mammals and the sampling number for each tick species are indicated in parentheses.

**Figure 2 viruses-15-02353-f002:**
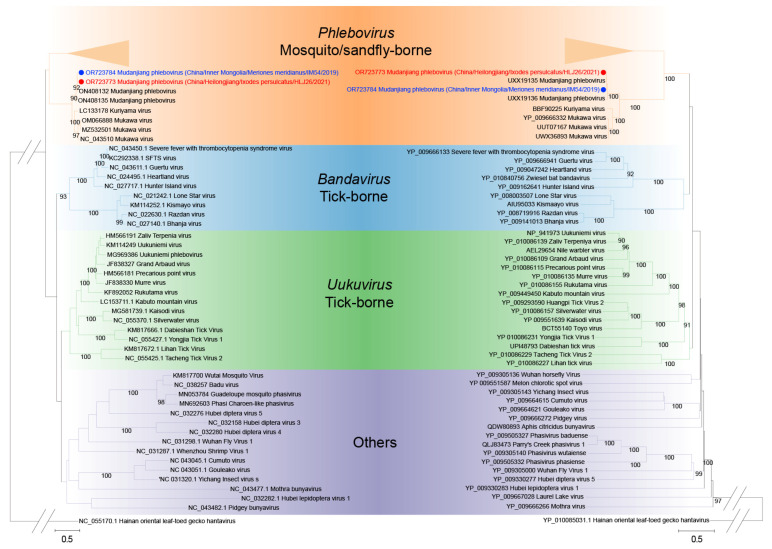
Phylogenetic analysis of the MJPV partial L segment. The maximum likelihood phylogenetic tree was constructed based on a 2844 bp fragment of the partial L segment. MJPV obtained from Ixodes persulcatus and from Meriones meridianus in the current study were, respectively, labeled with red and blue dots. The trees were generated using MEGA 7.0.26 and analyzed with 1000 bootstrap replicates.

**Figure 3 viruses-15-02353-f003:**
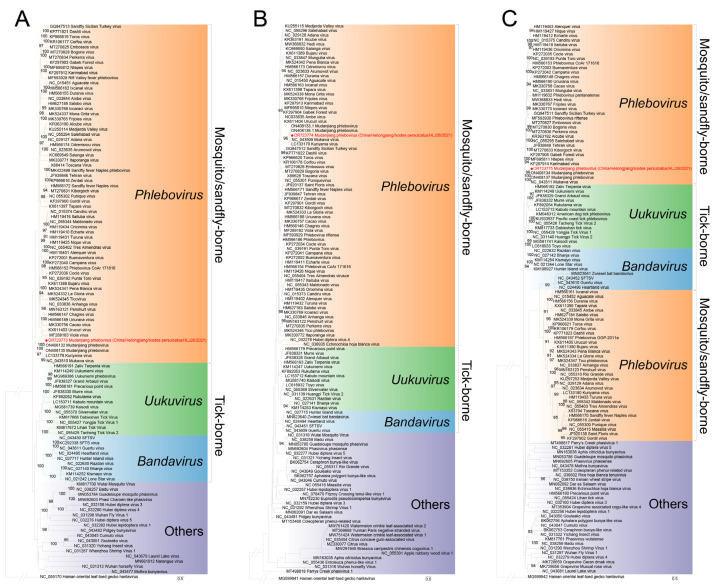
Phylogenetic analysis of MJPV full RNA segments. (**A**) The maximum likelihood phylogenetic tree constructed based on the complete nucleotide sequences of the L segment. (**B**) The maximum likelihood phylogenetic tree constructed based on the complete nucleotide sequences of the M segment. (**C**) The maximum likelihood phylogenetic tree constructed based on the complete nucleotide sequences of the S segment. The MJPV obtained in the current study is denoted by a red dot. The trees were generated using MEGA 7.0.26 and analyzed with 1000 bootstrap replicates.

**Figure 4 viruses-15-02353-f004:**
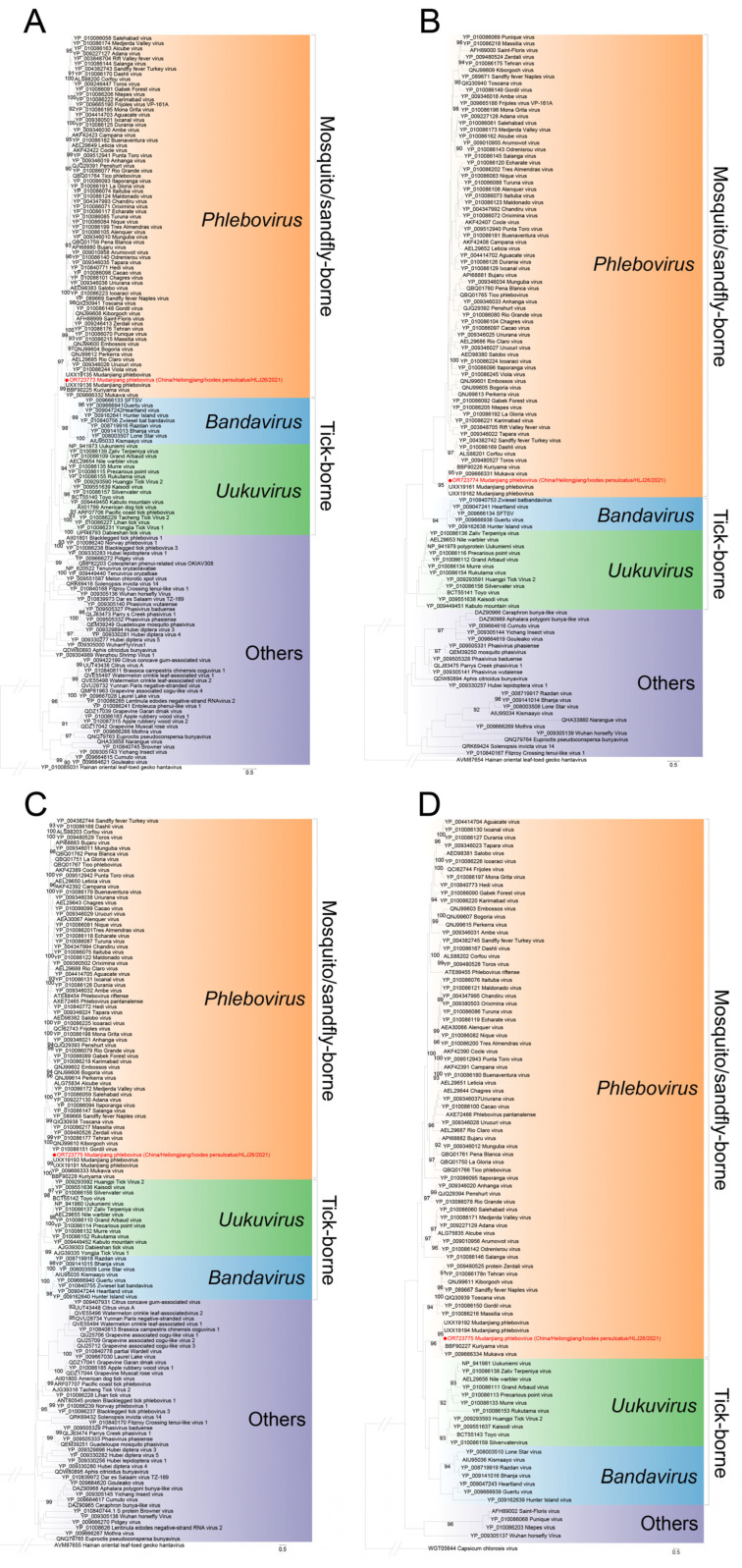
Phylogenetic analysis of MJPV for the deduced amino acid sequences. (**A**) The maximum likelihood phylogenetic tree based on the amino acid sequences of the RNA-dependent RNA polymerase. (**B**) The maximum likelihood phylogenetic tree based on the amino acid sequences of the glycoprotein precursor. (**C**) The maximum likelihood phylogenetic tree based on the amino acid sequences of nucleoprotein protein. (**D**) The maximum likelihood phylogenetic tree based on the amino acid sequences of nonstructural protein. The current MJPV is labeled with a red dot. The trees were generated using MEGA 7.0.26 and analyzed with 1000 bootstrap replicates.

**Table 1 viruses-15-02353-t001:** Primers used in the RT-PCR for Mudanjiang phlebovirus detection.

Primer Set	Primer	Sequence (5′→3′)	Methods
ppL1 *	HRT-GL2759F	CAGCATGGIGGIYTIAGRGAAATYTATGT	RT-PCR based sequencing
	HRT-GL3276R	GAWGTRWARTGCAGGATICCYTGCATCAT	
ppL2 *	TBPVL2759F	CAGCATGGIGGICTIAGAGAGAT	RT-PCR based sequencing
	TBPVL3267R	TGIAGIATSCCYTGCATCAT	

* The primers were derived from reference [14].

**Table 2 viruses-15-02353-t002:** Tick and wild small mammals screened for Mudanjiang phlebovirus.

Tick and Wild Small Mammals	Location	Year	Species (*n*)	No. Total Tested	No. (%) of Positive
Tick	Total			1815	4 (0.22)
	Heilongjiang	2019, 2021	*Ixodes persulcatus* (665), *Haemaphysalis concinna* (45)	710	4 (0.56)
	Inner Mongolia	2019	*Dermacentor silvarum* (336)	336	0 (0)
	Henan	2019	*Haemaphysalis longicornis* (109)	109	0 (0)
	Shanxi	2019	*Haemaphysalis longicornis* (144)	144	0 (0)
	Liaoning	2019	*Haemaphysalis longicornis* (174)	174	0 (0)
	Shandong	2019	*Haemaphysalis longicornis* (172)	172	0 (0)
	Zhejiang	2018	*Rhipicephalus microplus* (170)	170	0 (0)
Wild small mammals	Total			805	1 (0.12)
	Heilongjiang	2017	*Microtus fortis* (5), Rattus *norvegicus* (22), *Apodemus agrarius* (16), *Tamias sibiricus* (1), *Mus musculus* (1)	45	0 (0)
	Inner Mongolia	2019	*Spermophilus dauricus* (35), *Allocricetulus eversmanni* (4), *Cricetulus barabensis* (2), *Cricetulus migratorius* (38), *Phodopus sungorus* (6), *Dipus sagitta* (5), *Allactaga sibirica* (28), *Mus musculus* (2), *Phodopus roborovskii* (4), *Meriones unguiculatus* (183), *Meriones meridianus* (64), *Lasiopodomys mandnarinus* (15)	386	1 (0.26)
	Henan	2018	*Apodemus agrarius* (15), *Niviventer confucianus* (1), *Berylmys bowersi* (2), *Rattus norvegicus* (5)	23	0 (0)
	Xinjiang	2018	*Meriones tamariscinus* (3), *Spermophilus erythrogenys* (31), *Eothenomys miletus* (1), *Rhombomys opimus* (77), *Sundamys muelleri* (4), *Lasiopodomys oeconomus* (1), *Rattus norvegicus* (20), *Meriones libycus* (48), *Vormela peregusna* (1), *Cricetulus migratorius* (5), *Mustela nivalis* (1), *Sorex araneus Linnaeus* (2), *Microtus kikuchii* (1), *Allactaga sibirica* (8), *Marmota himalayana* (1), *Mus musculus* (104), *Apodemus sylvaticus* (15), *Spermophilus parryii* (25), *Meriones meridianus* (3)	351	0 (0)

## Data Availability

Data are contained within the article and Appendix A.

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
