# Peer review of "Identification of Mudanjiang Phlebovirus in the Daxing’anling Region of China"

_viruses, 2023, doi:10.3390/v15122353_

Round 1

Reviewer 1 Report

Comments and Suggestions for Authors

The study had a good sample and presents important eco-epidemiological data for the region. 

However, the presentation of the materials and methods seems confusing to me in some points, and I suggest some minor revisions on the text, to make it clearer for the reader:

Line 82: "Organ specimens such as heart, lung, liver, kidney and spleen of wild small mammals were obtained in a BSL-2 laboratory... All samples were stored in tubes and placed in liquid nitrogen upon capture, transferred to the laboratory and stored at 80°C for later use." 

All samples mean only the ticks, or are the animals also considered? Or the animals were transported alive to the laboratory, and then euthanized and tissues collected at BSL-2?

___________

Line 97: "Seventy-five tick samples belonging to five species .... along with twelve spleen samples belonging to four dominant wild small animal species ... underwent metagenomic analysis...". 

Following the text...

Line 103 "A one-step RT-PCR system was employed to test the extracted RNAs..."
I'm assuming that these are extracted samples that went to NGS, because at no point in the text are other extractions mentioned. 

Following the text...

Line 144: "A total of 1815 adult ticks, comprised five tick species, and 805 wild small mammals consisting of 37 rodent species, collected from September 2017 to October 2021 were tested for the presence of MJPV"

Here we see that a much broader N was tested by RT-PCR, but where is the description of this N in the materials on methods? I suggest describing it next to item 2.3 or maybe 2.2.

Reviewer 2 Report

Comments and Suggestions for Authors

All comments are in the text.  In general, a good survey of a virus in ticks and animals. Wordy text could be made more concise.

Comments on the Quality of English Language

Wordy text could be made more concise.

Author Response

Wordy text could be made more concise.

Re: We appreciate your important comment on how to improve our manuscript. Done as suggested.

Reviewer 3 Report

Comments and Suggestions for Authors

Fang et al. reported the detection and identification of Mudanjiang phlebovirus in Daxing’anling region. The virus is a newly tick-borne phlebovirus. This study found it could be detected in I. persulcatus ticks and Merinoes meridianus rodents, suggesting the virus circulates in a wide range of animals. Generally, this is a very concise story and the manuscript is well writen. I only have a few suggestions:

Line 104: Please explain HRT.

Line 117: Phenuiviridae should be in italics.

Line 163: I can’t find Table 2

Lines 209-210: Please provide a reference for the sentence.
